# In Silico Probiogenomic Characterization of *Lactobacillus delbrueckii* subsp. *lactis* A4 Strain Isolated from an Armenian Honeybee Gut

**DOI:** 10.3390/insects14060540

**Published:** 2023-06-09

**Authors:** Inga Bazukyan, Dimitrina Georgieva-Miteva, Tsvetelina Velikova, Svetoslav G. Dimov

**Affiliations:** 1Faculty of Biology, Yerevan State University, Yerevan 0025, Armenia; bazukyan@ysu.am; 2Faculty of Biology, Sofia University St. Kliment Ohridski, 8 Dragan Tzankov Str., 1164 Sofia, Bulgaria; d.georgieva@biofac.uni-sofia.bg; 3Medical Faculty, Sofia University St. Kliment Ohridski, 1 Kozyak Str., 1407 Sofia, Bulgaria; tsvelikova@medfac.mu-sofia.bg

**Keywords:** honeybees, endosymbionts, probiotics, probiogenomic analysis, whole-genome sequencing, *L. delbrueckii* ssp. *lactis*

## Abstract

**Simple Summary:**

Although *L. delbrueckii* strains have been isolated from different bee products and beehive niches to our knowledge, this is the first report of the isolation of the *L. delbrueckii* strain from the gut of a honeybee. Furthermore, the genomic analysis of the strain revealed two major aspects of its genomic constitution: (1) it possesses many genetic determinants attributing probiotic properties for the honeybees, and (2) it shows a reduction in size which is a typical adaptation of bacteria undergoing transformation to an endosymbiont. These two observations motivated us to hypothesize that in the case of *L. delbrueckii* subsp. *lactis* A4, we witness the first case of an *L. delbrueckii* strain evolution to a honeybee endosymbiont.

**Abstract:**

A *Lactobacillus delbrueckii* ssp. *lactis* strain named A4, isolated from the gut of an Armenian honeybee, was subjected to a probiogenomic characterization because of its unusual origin. A whole-genome sequencing was performed, and the bioinformatic analysis of its genome revealed a reduction in the genome size and the number of the genes—a process typical for the adaptation to endosymbiotic conditions. Further analysis of the genome revealed that *Lactobacillus delbrueckii* ssp. *lactis* strain named A4 could play the role of a probiotic endosymbiont because of the presence of intact genetic sequences determining antioxidant properties, exopolysaccharides synthesis, adhesion properties, and biofilm formation, as well as an antagonistic activity against some pathogens which is not due to pH or bacteriocins production. Additionally, the genomic analysis revealed significant potential for stress tolerance, such as extreme pH, osmotic stress, and high temperature. To our knowledge, this is the first report of a potentially endosymbiotic *Lactobacillus delbrueckii* ssp. *lactis* strain adapted to and playing beneficial roles for its host.

## 1. Introduction

It is not easy to estimate precisely the economic value of honeybees as pollinators of crops and other plants. However, in any case, worldwide, it ranges up to dozens of billions of USD annually [1]. One of the main factors affecting the honeybee’s welfare is the composition of their gastrointestinal microbiota, which is well characterized and comprises several phylotypes [2]. A crucial role in the honeybee’s welfare is played by several core species and genera such as *Snodgrassella alvi*, *Gilliamella apicola*, *Frischella perrara*, and *Commensalibacter sp.* but also members of the genus *Bifidobacterium* and the former genus *Lactobacillus* [3]. Their representatives play crucial roles in pollen degradation and sugar breakdown [4,5] but also in the protective biofilm formation [6], as well as playing a defensive role by stimulating the immune response [7].

Members of the former genus *Lactobacillus* (before their re-classification in 2020 [8]), being part of the core microbiota, among other things, such as the breakdown of plant sugars, are reported to stimulate the host immune system, to protect against pathogens [9]. However, they have also been reported recently to modulate host learning and memory behaviors via regulating tryptophan metabolism [10]. Some of the most often isolated strains belong to *Lactiplantibacillus plantarum*, *Lactiplantibacillus pentosus*, *Limosilactobacillus fermentum*, *Lacticaseibacillus paracasei*, and *Levilactobacillus brevis* [11,12].

Till now, *Lactobacillus delbrueckii* isolates have not been reported to be present within the honeybee intestinal microbiota but have been reported to be present within the bee bread in hives in the Anatolia region of Turkey [13]. This lack of information about the presence and the role of *Lactobacillus delbrueckii* within the gastrointestinal microbiota motivated the probiogenomic [14] study of *Lactobacillus delbrueckii* subsp. *lactis* strain A4 was isolated from a honeybee gut in 2017 from a worker bee in the town of Artashat in the Republic of Armenia. This strain was first selected because of its antagonistic activity against some pathogens, such as *Escherichia coli* and *Salmonella typhimurium*, as well as because of its ability to grow at a wide range of pH (4–9) and to assimilate arabinose as a carbon source, characteristics suggesting probiotic potential.

## 2. Materials and Methods

### 2.1. Sampling and Maintenance of the Strain

The sampling was carried out in an apiary situated in the Ararat province in the town of Artashat (40°24′18″N, 44°34′35″E). The strain A4 was isolated from the gut of a young worker honeybee sampled on 25.08.2017 by enrichment of 10% skim milk with the particles of the honeybee’s gut. First, 50 µL from the inoculated milk was mixed with 1 mL of sterile peptone water (PW). Next, 10-fold serial dilutions in PW were made up to 10–4, then 100 µL of the 10^−2^ and 10^−3^ dilutions were plated on Petri dishes with De Man, Rogosa, and Sharpe (MRS) 1.5% agar (Merck, Darmstadt, Germany). After incubation for 36–48 h at 30 °C, single colonies were picked up and inoculated in 3 mL MRS liquid broth and incubated at 30 °C for 24 h. After that period, ten isolates were plated on MRS 1.5% agar with a sterile loop by the agar streaking method and incubated at 30 °C for 24 h. This procedure was repeated thrice, and the cultures were microscopically monitored for contamination. Finally, stocks of skim milk were prepared from each isolate [15] and were kept at −70 °C.

### 2.2. DNA Isolation

Total DNA was isolated from 3mL 24-h liquid culture in MRS broth obtained by single colony inoculation. The DNA was extracted using the “Gram Plus & Yeast Genomic DNA Purification Kit” (EURx, Gdansk, Poland) according to the manufacturer’s instructions. The final elution was performed in 70 µL of the kit’s elution buffer. The quality of the isolated DNA was monitored on a 0.8% agarose gel in a TBE buffer system, while the concentration was determined on a Quantus™ fluorimeter (Promega, Madison, WI 53711 USA). The isolated DNA was further stored at −70 °C.

### 2.3. DNA Techniques

First, for preliminary species determination, total DNA isolated from the strain was sent to Macrogen (Seoul, Republic of Korea) for 16S ribosomal RNA gene sequencing by the Sanger chain termination method with the primers pair 27F/1492R [16] on the ABI 3730xl System. 50 µL of the isolated DNA was shipped on dry ice to BGI (Tai Po, N.T. Hong Kong) for whole-genome sequencing on the DNBseq platform [17] for DNB-SEQ PE100/PE150 sequencing with a generation of 1 Gb of data.

### 2.4. Bioinformatic Analyses

The forward and the reverse reads of the 16S rRNA gene sequencing were assembled with SeqMan™ II v.5.01 software (DNASTAR Inc., Madison, WI 53705, USA), and the assembled contig was uploaded to the GenBank of the NCBI. The results were analyzed online at the NCBI by the Nucleotide BLAST program (https://blast.ncbi.nlm.nih.gov/Blast.cgi, accessed on 1 February 2023) [18]. The quality of the raw data obtained from the DBBseq platform was monitored by the online tool FastQC v. 0.11.9 [19]. The contigs and scaffolds were assembled by the online assembler SPAdes v. 3.15.4 [20]. The quality of the assembled contigs was assessed by the online tool Quast v. 5.2.0 [21]. The contigs shorter than 200 bp were filtered by the Filter FASTA v. 1.9.1.0 [22] online software. The last four steps were performed on the Galaxy server (usegalaxy.eu, accessed on 3 February 2023). To confirm the species attribution, the average nucleotide identity of the assembled genome was evaluated with the ANI calculator [23] located on the EzBioCloud (ezbiocloud.net/tools/ani, accessed on 11 February 2023). The assembled contigs were uploaded to the GenBank of the NCBI. The annotation was performed by the NCBI Prokaryotic Genome Annotation Pipeline (PGAP) (https://www.ncbi.nlm.nih.gov/genome/annotation_prok/, accessed on 11 February 2023) [24,25,26]. The initial genotypic characterization was performed on the Center for Genomic Epidemiology (genomicepidemiology.org/services/, accessed on 11 February 2023) using the following tools: PlasmidFinder 2.1 (database version: 18 January 2023) [27,28,29] for the presence of plasmid replicons; ResFinder 4.1 (database version: 24 May 2022 and ResFinderFG 2.0 (database version: 30 June 2022 [27,29,30,31] for the presence of antibiotics resistance genetic determinants; PathogenFinder 1.1 [27,32] for the presence of pathogenicity genetic determinants; and MobileElementFinder v. 1.0.3 (database version: 9 June 2020) [33] for identification of mobile genetic elements which could be related to antimicrobial resistance genes and virulence factors. The presence of bacteriocins’ genetic determinants was checked with the online tool BAGEL4 (http://bagel4.molgenrug.nl/, accessed on 11 February 2023) [34]. The presence of the genetic determinants determining potential probiotic properties was performed using the NCBI’s Sequence Set Browser (https://www.ncbi.nlm.nih.gov/Traces/wgs/, accessed on 15 February 2023), followed by BLAST analyses against the NCBI’s database.

## 3. Results

### 3.1. Genome Assembly, Annotation, and Characterization

The results from sequencing the 16S rRNA gene of isolate A4 are available at the NCBI’s GenBank under the accession number OP784418.1. From the raw data of the next-generation whole-genome shotgun sequencing, 145 contigs were assembled. The assembly statistics are summarized in Table 1. The assembled contigs of the draft genome are available at the NCBI’s GenBank under accession number JAQSVE000000000. Both BLAST of the 16S rRNA gene sequence and ANI of the assembled genome attributed the isolate to *Lactobacillus delbrueckii* ssp. *lactis* based on the 16S rRNA gene homology and the average nucleotide identity, respectively, compared with the NCBI databases. The results from the annotation are presented in Table 2. No plasmid origins of replication were detected by the PlasmidFinder tool. Neither were found genetic determinants for antibiotic resistance and factors of pathogenicity by the ResFinder, ResFinderFG, and PathogenFinder tools. The MobileElementFinder tool revealed the presence of an intact ISL6 insertion sequence belonging to the IS3 family (IS150 group) and seventeen 3′- or 5′- truncated mobile elements belonging to the ISLre2, ISL3, IS4, IS30, IS110, and IS256 families.

### 3.2. Genotypic Characterization of L. delbrueckii A4

The results of the BLAST analysis of the genes with the potential to determine probiotic properties are presented in Table 3. All the listed database hits have the same length with the query sequence, a query cover of 100%, and a percentage of identity of 100.00%.

## 4. Discussion

The analysis of the draft genome assembly performed with the Quast tool (Table 1) revealed that the assembly quality has the potential to guarantee correct analysis of *L. delbrueckii* ssp. *lactis* A4 genome, mainly because of the number of the contigs greater than 1000 bp, the N50, N90, L50, and L90 values. The calculated genome size based on all assembled contigs was 1,881,475 bp. This value was not significantly different from the values of the genome sizes calculated based on the total length and the contigs larger than 1000 bp, which once again confirmed the good quality of the assembly. The calculated GC content was 49.84%—a typical value for the species [35]. Surprisingly, when it was compared to other *L. delbrueckii* ssp. *lactis* strains, the A4 stood out with its clearly reduced genome size (Table 4).

More interesting results were obtained from the annotation of the genome, which revealed a reduction not only in the genome size but also in the number of genes. *L. delbrueckii* ssp. *lactis* A4 possesses the least total number of genes when compared with the other 14 randomly chosen strains of the same species, the least number of pseudogenes, and, with one exception, the least number of RNA coding genes. This tendency is also partially visible when comparing the numbers of the protein-coding genes; still, four other strains possess a lesser number.

It is quite difficult to speculate why *L. delbrueckii* ssp. *lactis* A4 possesses a reduced genome. However, considering its unusual origin (to our knowledge, *L. delbrueckii* was never reported to be isolated from a honeybee gut), the hypothesis of some specialization to the honeybee gut ecological niche could not be excluded. It is observed and documented that the symbiosis between insects and bacteria leads to mutual adaptations. One aspect of these symbiotic interactions is the reduction of the genome size of the symbiotic bacteria [36,37]. This phenomenon also occurs in honeybees, as documented for *Apibacter sp.* [38], and the obligate endosymbionts *Gillamella apicola* and *Snodgrassella alvi* [39]. The distinctiveness of the *L. delbrueckii* ssp. *lactis* A4, as an example of a strain undergoing such adaptation, is further supported by the fact that, till now, to our knowledge, this is the only representative of the species isolated from a honeybee gut microbiota. We believe that this hypothesis is also supported by the fact that the reduction in the number of the genes affects mainly the RNA coding genes and pseudogenes, but to a minor extent, the number of the protein-coding genes, and thus keeping its “metabolic potential” which in the case of the species *L. delbrueckii* which lacks virulence factors and pathogenicity determinants, is beneficial for the host.

Even though *lactobacilli* (in the broader meaning of the former big genus before its reclassification in 2020 [8]) are considered a part of the honeybees’ core endosymbionts [3], till now, *L. delbrueckii* strains have not been reported to be isolated from the honeybees. So, logically, if the hypothesis of the genome adaption is true, what would be the role of *L. delbrueckii* ssp. *lactis* A4 for the honeybees, so is it selectively maintained long enough for genomic changes to occur? A logical answer to this question is that it possesses some crucial probiotic properties for the host. So, the next step of this research was to make a probiogenomic characterization, a notion proposed by de Jesus et al. to describe the analysis of the genome for the presence of genetic determinants giving probiotic properties [14], which in honeybees could be mainly antioxidant properties, exopolysaccharides synthesis, stress tolerance, adhesion to the gut epithelium and biofilm formation, as well as antagonistic activity towards pathogens.

It is documented that in human and animal models, lactic acid bacteria (LAB) and particularly *L. delbrueckii,* exert a beneficial protective effect against reactive oxygen species (ROS) on the intestinal mucosa and epithelium [40]. The studies of some of the mechanisms determining these antioxidant activities in LAB and *lactobacilli* revealed that they involve the synthesis of several enzymes and proteins [41,42], as well as the synthesis of exopolysaccharides (EPS) [43]. Nineteen enzymes and proteins involved in the common oxidative stress resistance are listed in different *Lactobacillus* species [41]. Genetic determinants for 7 of them were found in *L. delbrueckii* ssp. *lactis* A4: NAD(P)H-dependent oxidoreductase, DNA starvation/stationary phase protection protein, DsbA family protein, NAD(P)/FAD-dependent oxidoreductase, thioredoxin family protein, thioredoxin-disulfide reductase and thioredoxin (Table 3). Oxidative resistance genes are rather scarce among *lactobacilli* [41], so finding such a number in only one strain should not be considered a coincidence.

Additional antioxidant activity can be attributed to the chelator activity of the EPS, which can sequester heavy metal ions [43]; however, they can exert additional beneficial properties on their own. Different gene products participate in exopolysaccharide synthesis [44,45,46]. Genetic determinants for 9 of them were found in *L. delbrueckii* ssp. *lactis* A4 genome: exopolysaccharide biosynthesis protein, NADP-dependent phosphogluconate dehydrogenase, phosphoglucosamine mutase, beta-phosphoglucomutase, UDP-glucose-hexose-1-phosphate uridylyltransferase, UTP-glucose-1-phosphate uridylyltransferase GalU, bifunctional UDP-N-acetylglucosamine, diphosphorylase/glucosamine-1-phosphate N-acetyltransferase GlmU, oligosaccharide flippase family protein and flippase (Table 3). They strongly suggest an ability of EPS synthesis, which in turn can contribute to the already mentioned protection from heavy metals, but also by interfering with the adhesion of pathogens, as well as by exerting immunomodulatory properties and helping the biofilm formation [46].

Despite being related to EPS production, adhesion, and biofilm formation, they depend on different genetic determinants [47,48,49,50]. Eight genes encoding SLAP domain-containing proteins, D-alanyl-lipoteichoic acid biosynthesis protein DltB, D-alanyl-lipoteichoic acid biosynthesis protein DltD, elongation factor Tu, type I glyceraldehyde-3-phosphate dehydrogenase and triose-phosphate isomerase were found within the *L. delbrueckii* ssp. *lactis* A4 genome (Table 3). They could determine good intestinal adhesion properties, which in turn can result in competitiveness with pathogens [48] but also in immunomodulating and anti-inflammatory effects, as shown in in vitro studies [47]. Additionally, the strain probably possesses the ability for biofilm formation because three genes determine the synthesis of AI-2E family transporters and cell wall metabolism sensor histidine kinase WalK, involved in signaling and quorum sensing mechanisms [49,51,52,53].

The inhibition of pathogens’ growth could also be considered a probiotic property. In general, this property is attributed to the synthesis of bacteriocins or low molecular weight acids, which decrease the pH to levels intolerable to most pathogenic species. However, this is obviously not the case for the A4 strain because of the lack of bacteriocin genetic determinants and because it expresses this ability in neutralized conditions. Two alternatives are the production of H_2_O_2_ and biogenic amines. Lactate oxidase and pyruvate oxidase are two enzymes whose activity can lead to the production of H_2_O_2_ [54,55], and their genetic determinants were found within the A4 genome. Furthermore, genes encoding eight enzymes participating in the amino acids metabolism, which could also be involved in biogenic amines synthesis [56], were found: pyridoxal-dependent decarboxylase, orotidine-5’-phosphate decarboxylase, carboxymuconolactone decarboxylase family protein, diphosphomevalonate decarboxylase, diaminopimelate decarboxylase, putative ornithine decarboxylase, and bifunctional phosphopantothenoylcysteine decarboxylase/phosphopantothenate-cysteine ligase CoaBC (Table 3). However, for the moment, we do not have enough data on which of those two mechanisms is involved in the inhibitory activity, if it is due to both of them or some other unknown mechanism.

Finally, a probiotic strain to survive and be selectively maintained within the gut microbiome, especially in the case of the honeybees, should be stress tolerant and resistant to acidic and alkaline pH, osmotic stress, and heat. Extreme pH values are found in the different parts of the honeybee’s gut, while the feeding could achieve osmotic stress with honey. In addition, the body temperature could sometimes rise dramatically during a flight on a warm sunny day. Many genes are involved in coping with these extrema [14], and some were found within the *L. delbrueckii* ssp. *lactis* A4 genome—those encoding orotidine-5’-phosphate decarboxylase, S-ribosylhomocysteine lyase, phosphopyruvate hydratase, several AAA family ATPases, Na + /H+ antiporter NhaC, two-component system regulatory protein YycI, nucleotide exchange factor GrpE, molecular chaperones DnaK and DnaJ, heat-inducible transcriptional repressor HrcA, chaperonin GroEL, oligoendopeptidase F, CTP synthase, glucosamine-6-phosphate deaminase, aquaporin family proteins, aldo/keto reductase, ATP-dependent Clp protease ATP-binding subunit ClpX and F0F1 ATP synthase subunit alpha, beta, gamma, epsilon, B and C (Table 3). These findings suggest strong stress tolerance against the listed factors.

## 5. Conclusions

The whole-genome sequencing of *L. delbrueckii* ssp. *lactis* A4 genome suggests that this strain is an unusual one, probably undergoing adaptation to the honeybee gut conditions and thus becoming part of the symbiont microflora—a process never documented before for this species. The probiogenomic bioinformatic analysis supports this hypothesis because it revealed genetic determinants for a different probiotic for the host properties. The strain is stress-tolerant—a prerequisite to survive and to be selectively maintained within the insect’s microbiota.

## Figures and Tables

**Table 1 insects-14-00540-t001:** Statistical results of *L. delbrueckii* ssp. *lactis* A4 genome’s assembly.

Statistics Parameter	Value
Number of contigs	114
Number of contigs (>0 bp)	145
Number of contigs (>1000 bp)	106
Largest contig	114,871
Total length	1,871,570
Total length (>0 bp)	1,881,475
Total length (>1000 bp)	1,866,007
N50	32,562
N90	8504
auN	41,377
L50	17
L90	61
GC (%)	49.84
* **Mismatches** *	
Number of N’s per 100 kbp	0
Number of N’s	0

**Table 2 insects-14-00540-t002:** Results from the annotation of the *L. delbrueckii* ssp. *lactis* A4 genome.

Genomic Annotation Parameter	Value
Genes (total)	1917
CDSs (total)	1854
Genes (coding)	1774
CDSs (with protein)	1774
Genes (RNA)	63
rRNAs	1, 2, 2 (5S, 16S, 23S)
complete rRNAs	1 (5S)
partial rRNAs	2, 2 (16S, 23S)
tRNAs	55
ncRNAs	3
Pseudo Genes (total)	80
CDSs (without protein)	80
Pseudo Genes (ambiguous residues)	0 of 80
Pseudo Genes (frameshifted)	38 of 80
Pseudo Genes (incomplete)	38 of 80
Pseudo Genes (internal stop)	18 of 80
Pseudo Genes (multiple problems)	12 of 80
CRISPR Arrays	1

**Table 3 insects-14-00540-t003:** Results from the BLAST analysis of the gene products potentially determining probiotic properties of *L. delbrueckii* ssp. *lactis* A4.

Gene Products with the Potential to Determine Probiotic Properties	GeneBank Accession Number	BLAST hit Accession Number	Length (Number of Amino Acids)	Max Score/Total Score	E Value
**Exopolysaccharides synthesis**					
exopolysaccharide biosynthesis protein	MDD1332515.1	WP_002879977.1	258	538	0.0
NADP-dependent phosphogluconate dehydrogenase	MDD1332818.1	WP_003616396.1	461	954	0.0
phosphoglucosamine mutase	MDD1331320.1	WP_003617321.1	450	920	0.0
beta-phosphoglucomutase	MDD1331831.1	WP_003616652.1	221	440	2 × 10^−155^
UDP-glucose-hexose-1-phosphate uridylyltransferase	MDD1331739.1	WP_273963523.1	488	1007	0.0
UTP-glucose-1-phosphate uridylyltransferase GalU	MDD1331545.1	WP_231539737.1	304	622	0.0
bifunctional UDP-N-acetylglucosamine diphosphorylase/glucosamine-1-phosphate N-acetyltransferase GlmU	MDD1332146.1	WP_120490223.1	461	944	0.0
oligosaccharide flippase family protein	MDD1331971.1	WP_016395806.1	475	957	0.0
oligosaccharide flippase family protein	MDD1332322.1	WP_273964740.1	477	955	0.0
flippase	MDD1332932.1	WP_273964869.1	476	967	0.0
**Adhesion**					
SLAP domain-containing protein	MDD1332405.1	WP_273964748.1	536	1082	0.0
SLAP domain-containing protein	MDD1332404.1	WP_120490196.1	399	794	0.0
SLAP domain-containing protein	MDD1331713.1	WP_191669459.1	178	362	6 × 10^−126^
D-alanyl-lipoteichoic acid biosynthesis protein DltB	MDD1332807.1	WP_003613651.1	410	837	0.0
D-alanyl-lipoteichoic acid biosynthesis protein DltD	MDD1332805.1	WP_273964847.1	428	882	0.0
elongation factor Tu	MDD1331613.1	WP_003617518.1	396	797	0.0
type I glyceraldehyde-3-phosphate dehydrogenase	MDD1331303.1	WP_002879985.1	338	701	0.0
triose-phosphate isomerase	MDD1331305.1	WP_002879988.1	252	518	0.0
**H_2_O_2_ production**					
lactate oxidase	MDD1332690.1	WP_231520111.1	408	847	0.0
pyruvate oxidase	MDD1332991.1	WP_273964886.1	607	1253	0.0
**Antioxidant properties**					
NAD(P)H-dependent oxidoreductase	MDD1332108.1	WP_002879531.1	179	366	9 × 10^−128^
NAD(P)H-dependent oxidoreductase	MDD1332107.1	WP_231533908.1	182	371	1 × 10^−129^
DNA starvation/stationary phase protection protein	MDD1331912.1	WP_002878317.1	155	321	1 × 10^−110^
DsbA family protein	MDD1331257.1	WP_086356203.1	214	437	9 × 10^−155^
NAD(P)/FAD-dependent oxidoreductase	MDD1331908.1	WP_013440359.1	444	914	0.0
NAD(P)/FAD-dependent oxidoreductase	MDD1331702.1	WP_236155003.1	443	905	0.0
thioredoxin family protein	MDD1332832.1	WP_002879808.1	106	217	5 × 10^−71^
thioredoxin-disulfide reductase	MDD1331288.1	WP_002879964.1	310	633	0.0
thioredoxin	MDD1332291.1	WP_016396729.1	103	213	1 × 10^−69^
**Biofilm formation**					
AI-2E family transporter	MDD1332686.1	WP_016396259.1	357	721	0.0
AI-2E family transporter	MDD1331996.1	WP_273964107.1	377	748	0.0
AI-2E family transporter	MDD1331266.1	WP_130137291.1	393	794	0.0
cell wall metabolism sensor histidine kinase WalK	MDD1332604.1	WP_041811526.1	631	1294	0.0
**Biogenic amines production**					
pyridoxal-dependent decarboxylase	MDD1332960.1	WP_260267444.1	108	227	7 × 10^−75^
orotidine-5’-phosphate decarboxylase	MDD1332839.1	WP_013439932.1	240	499	3 × 10^−178^
carboxymuconolactone decarboxylase family protein	MDD1332744.1	GHN63519.1	106	219	1 × 10^−71^
diphosphomevalonate decarboxylase	MDD1332354.1	WP_016396687.1	319	661	0.0
diaminopimelate decarboxylase	MDD1332065.1	WP_231540913.1	436	900	0.0
putative ornithine decarboxylase	MDD1331554.1	WP_231540563.1	695	1439	0.0
bifunctional phosphopantothenoylcysteine decarboxylase/phosphopantothenate-cysteine ligase CoaBC	MDD1331448.1	WP_138463443.1	399	801	0.0
**Stress** **tolerance**					
orotidine-5’-phosphate decarboxylase	MDD1332839.1	WP_013439932.1	240	499	3 × 10^−178^
S-ribosylhomocysteine lyase	MDD1332544.1	WP_002879583.1	159	329	1 × 10^−113^
phosphopyruvate hydratase	MDD1332002.1	WP_002879863.1	425	873	0.0
AAA family ATPase	MDD1332910.1	WP_273964864.1	507	1039	0.0
AAA family ATPase	MDD1332713.1	WP_231534278.1	808	1600	0.0
AAA family ATPase	MDD1332527.1	WP_002879481.1	180	370	2 × 10^−129^
AAA family ATPase	MDD1332101.1	WP_130137762.1	259	523	0.0
AAA family ATPase	MDD1331784.1	WP_273963672.1	560	1146	0.0
AAA family ATPase	MDD1331771.1	WP_273963644.1	311	651	0.0
AAA family ATPase	MDD1331533.1	WP_035162369.1	731	1491	0.0
Na + /H+ antiporter NhaC	MDD1332823.1	WP_013440093.1	459	899	0.0
two-component system regulatory protein YycI	MDD1332602.1	WP_013438940.1	266	540	0.0
nucleotide exchange factor GrpE	MDD1331986.1	WP_130137528.1	199	392	4 × 10^−137^
molecular chaperone DnaK	MDD1331987.1	WP_236161911.1	614	1232	0.0
molecular chaperone DnaJ	MDD1331988.1	WP_130137527.1	379	775	0.0
heat-inducible transcriptional repressor HrcA	MDD1331985.1	WP_191669578.1	347	707	0.0
chaperonin GroEL	MDD1332210.1	WP_120490554.1	537	1070	0.0
oligoendopeptidase F	MDD1332691.1	WP_273964826.1	600	1239	0.0
CTP synthase	MDD1332152.1	WP_130183263.1	539	1115	0.0
glucosamine-6-phosphate deaminase	MDD1331907.1	WP_013440358.1	234	481	2 × 10^−171^
aquaporin family protein	MDD1332650.1	WP_013440193.1	234	454	1 × 10^−160^
aquaporin family protein	MDD1332638.1	WP_273964812.1	240	488	5 × 10^−174^
aldo/keto reductase	MDD1332535.1	WP_273964794.1	285	586	0.0
ATP-dependent Clp protease ATP-binding subunit ClpX	MDD1331611.1	WP_003617523.1	417	845	0.0
F0F1 ATP synthase subunit gamma	MDD1331368.1	WP_035161843.1	320	656	0.0
F0F1 ATP synthase subunit epsilon	MDD1331370.1	WP_130137331.1	146	293	8 × 10^−100^
F0F1 ATP synthase subunit B	MDD1331365.1	WP_130137329.1	168	331	5 × 10^−114^
F0F1 ATP synthase subunit C	MDD1331364.1	WP_130137328.1	74	140	9 × 10^−42^
F0F1 ATP synthase subunit alpha	MDD1331367.1	WP_002880068.1	503	1015	0.0
F0F1 ATP synthase subunit beta	MDD1331369.1	WP_013439311.1	479	965	0.0

**Table 4 insects-14-00540-t004:** Some genome properties of *L. delbrueckii* ssp. *lactis* A4 compared to other strains belonging to this species.

*L. delbrueckii* ssp. *lactis* Strain	Origin	GenBank Accession Number	Genome Size	Number of Genes	Number of Protein-Coding Genes	RNA Coding Genes	Pseudogenes
**A4**	**Honeybee gut**	**JAQSVE000000000.1**	**1,881,475**	**1917**	**1774**	**63**	**80**
KCTC 3034	n/a	CP023139.1	2,237,608	2251	1978	124	233
NCIMB 702468	dairy product	JAJNTF000000000.1	1,871,713	2036	1718	131	96
DSM 20076	dairy product	JAJNTW000000000.1	1,894,640	2016	1720	134	162
NCIMB 702465	dairy product	JAJNTI000000000.1	1,957,698	2082	1781	116	185
FAM 24850	dairy product	JAJNTA000000000.1	1,939,496	2076	1774	115	187
CIRM BIA 1374	dairy product	JAJNTY000000000.1	1,918,408	2060	1754	100	206
NCIMB 700280	dairy product	JAJNTN010000000	1,979,897	2107	1808	120	179
CIDCA 133	raw milk	CP065513.1	2,127,785	2165	1942	128	95
FAM 21784	human microbiome	VBSS00000000.1	2,005,191	2112	1853	108	151
NWC_2_2	fermented food	CP031023.1	2,269,179	2331	1934	121	276
CRL581	cheese starter	ATBQ00000000.1	1,909,893	2025	1750	88	187
MAG_rmk202_ldel	starter culture	CP046131.1	2,166,765	2163	1868	125	170
KCTC 3035	n/a	CP018156.1	1,972,735	2006	1788	123	95
NBIMCC 8250	n/a	JAJQWQ000000000.1	1,931,209	2009	1809	58	142

## Data Availability

The results from sequencing the 16S rRNA gene of *L. delbrueckii* ssp. *lactis* A4 are available at GenBank under the accession number OP784418.1. The assembled contigs of the draft genome are available at the NCBI’s GenBank under accession number JAQSVE000000000.

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
