# Peer review of "In Silico Probiogenomic Characterization of Lactobacillus delbrueckii subsp. lactis A4 Strain Isolated from an Armenian Honeybee Gut"

_insects, 2023, doi:10.3390/insects14060540_

Round 1

Reviewer 1 Report

Dear Authors,

Your work is very  interesting.  

I have a few comments about your work.

1) In the Materials and Methods section, did you have a negative control when you inoculated the culture? And one more wish for the MM section: why did you use skim milk. A little explanation in the text would not hurt, since your article will be read not only by narrow specialists.

2) you have errors in the References:

Line 295 - You cited a preprint that has already been published in the journal https://journals.plos.org/plosone/article?id=10.1371/journal.pone.0273844 and also there are unknown characters in the title.

397 line - the title of the journal is given after doi;

402 line - unknown characters in the title of the article, should be: Functional analysis of luxS in the probiotic strain Lactobacillus rhamnosus GG reveals a central metabolic role important for growth and biofilm formation;

In some places you use doi:, in other places you use doi:https.

Author Response

Dear reviewer,

Thank you for the high esteem of our work, but before all thank you for your valuable remarks which were taken into consideration and greatly improved our work!

Concerning your remarks:

  1. In our opinion, a negative control is not necessary because the milk was used as a partially selective media for lactic acid bacteria. As the milk is sterile, and the inoculation with gut’s particles was performed in sterile laboratory conditions, there are no reasons to expect a growth within a non-inoculated sterile milk.
  2. Concerning the remark of the skimmed milk used as a conservation agent, our own experience working with it in the last is very satisfying. However, accordingly to your suggestion, we introduced a proper reference within the text.
  3. Regarding the references, we used a reference manager which automatically formatted the references. However, we made manual corrections as you proposed.
    On behalf of all the co-authors, I would like to express our gratitude for your positive contribution to our article.
    Best regards,

Svetoslav Dimov

Reviewer 2 Report

The authors of this submission performed whole genome sequence analysis of Lactobacillus delbrueckii subsp. lactis A4 strain and evaluate its probiotic properties.

I have some suggestions to improve quality of the paper.

May be the title of paper should be reformulated as In silico probiogenomic characterization…

Line 39. Please, write a short paragraph according the main honey bee microbiota phylotypes as Alpha-1, Alpha-2.1 etc.

Line 58. Any citation to support this statement?

Line 58-61. How the authors determined these properties?

Line 64. worker honeybee or foragers? Also, describe of used methodology for isolation of Lactobacillus delbrueckii subsp. lactis strain A4.

Line 65. This sentence should be placed in the beginning of the paragraph.

Line 69. What does MRS mean?

Line 70. Merck – town, country of manufacturer?

Line 79. Please, see the remark above!

Line 85. The 16S rRNA gene or gene fragment should be amplified firstly before sequencing.

Lime 98. Filter FASTA v. 1.9.1.0 – Please, provide an online link for this software!

Line 152. 1,881,475 – bp?

Line 170. What the authors mean by saying unusual origin?

Line 196. LAB - Lactic acid bacteria?

Line 233. The inhibition of pathogens – inhibition of pathogens growth or?

Author Response

Dear reviewer,

Thank you for the high esteem of our work, but before all thank you for your valuable remarks which were taken into consideration and greatly improved our work!

Concerning your remarks:

  1. We greatly appreciate your suggestion for the change in the title, so we changed it.
  2. We are thankful for the remark concerning the phylotypes, we included a short sentence with the appropriate reference.
  3. Concerning the remarks about the paragraph on lines 58-61, the experimental results are part of another publication where the results for the activities of many isolates belonging to different bacterial species are reported and compared together. Unfortunately, this publication is still under review. We focused our work on L. delbrueckii A4 because it is not a typical endosymbiont, and have never been reported before to be part of the endosymbiont microflora. This is also the reason for sequencing its genome which revealed the interesting properties we report here. The inhibitory activity was determined by the agar diffusion method, while the growth at different pH by the cultivation within media with this pH.

Line 64  - a young worker honeybee was sampled directly from within the hive. This fact also support our hypothesis that A4 is undergoing endosymbiont changes.

Line 65. Done.

Line 69. The whole name of the broth was included.

Line 70. Corrected.

Line 79. Corrected

Line 85. The appropriate changes were introduced in the text.

Lime 98. It is indicated in 99 that the analysis is performed on the galaxy server with the web address.

Line 152. We are grateful for pointing us our omission!

Line 170. L. delbrueckii to our knowledge was never reported to be isolated from a honeybee gut. This is stated in other parts of the text but we also included it here.

Line 196. We are grateful for pointing us our omission!

Line 233. We are grateful for pointing us our omission!

On behalf of all the co-authors, I would like to express our gratitude for your positive contribution to our article.

Best regards,

Svetoslav Dimov

Reviewer 3 Report

Thank you for your response. I have checked it.

Since the content is very specialized, it is very important to explain it in an easy-to-understand manner for readers.

If you could show the whole picture of this paper in one figure (graphical abstracts), it would be easier for other readers to understand.

There is no problem.

Author Response

Dear reviewer,

Thank you for your remarks. We explained the basic idea of the study within the “Simple summary” section at the beginning of the article. We would like also to thank you for your suggestion to include a graphical abstract which we made.

Best regards,

Svetoslav Dimov

Round 2

Reviewer 3 Report

Authors responded our questions adequately. No additional questions.